# Trends in skin cancer incidence in Songkhla, Southern Thailand, 1989–2020: A population-based study on the impact of geographic variation

**Suchaya Pajareeyaphan**[1]*, **Paramee Thongsuksai**[1], **Hutcha Sriplung**[2], **Wit Wichaidit**[2]

1 Department of Pathology, Faculty of Medicine, Prince of Songkla University, Hatyai, Songkhla, Thailand,
2 Department of Epidemiology, Faculty of Medicine, Prince of Songkla University, Hatyai, Songkhla, Thailand

* saisue.pa@gmail.com

## Abstract

### Background

The incidence trends of skin cancer are increasing across the world. However, data from Southeast Asian countries, including Thailand, are limited. Songkhla, a province in southern Thailand, has a predominant occupation and religion across different geographic areas which may influencing the incidence. This study aimed to assess the trends in skin cancer incidence in Songkhla according to age, calendar period, birth cohort, and geographical areas.

### Methods

The study included patients diagnosed with squamous cell carcinoma, basal cell carcinoma, and melanoma of the skin between 1989 and 2020, as recorded in the Songkhla population-based cancer registry. Geographic areas were classified into four categories with remaining groups: urban VS rural, Muslim VS Buddhist, fishing and farming VS other occupations, and rubber plantation VS other occupations. Age-standardized incidence rates (per 100,000 population) were calculated, and trend analyses were performed using Joinpoint regression and age-period-cohort analysis.

### Results

The incidence of skin cancer in men declined after 2001 with annual percentage change rates of −2.24%, while it remained stable among women. However, when stratified by geographic area, the incidence among women showed a decline after 2016 in some areas. Overall, incidence rates were higher in urban than in rural areas, lower in predominantly Muslim than Buddhist areas, and lower in rubber plantation

**Data availability statement:** We have included the numerical values used to generate the graphs in the Supporting information files. However, we are unable to provide more detailed data because the dataset was obtained from the Division of Digital Innovation and Data Analytics, Faculty of Medicine, Prince of Songkla University. Due to institutional and patient privacy restrictions, these data are not publicly available but may be made available from the corresponding author upon reasonable request and with permission from the Division of Digital Innovation and Data Analytics and the Human Research Ethics Committee, Faculty of Medicine, Prince of Songkla University. Data may also be requested directly from the Division of Digital Innovation and Data Analytics, Faculty of Medicine, Prince of Songkla University (Tel: +66 74 451108; Email: happy@dida.psu.ac.th; Website: https://dida.psu.ac.th) and the Human Research Ethics Unit, Faculty of Medicine, Prince of Songkla University (Tel: +66 74 451157, +66 74 451149; Email: medpsu.ec@gmail.com; Website: https://hrec.medicine.psu.ac.th).

**Funding:** The author(s) received no specific funding for this work.

**Competing interests:** The authors have declared that no competing interests exist.

**Abbreviations:** NMSC, nonmelanoma skin cancer; BCC, basal cell carcinoma; SCC, squamous cell carcinoma; ASR, age-standardized incidence rate; UV, Ultraviolet; APCC, annual percentage change; APC, age-period-cohort; RR, rate ratio; 95% CI, 95% confidence intervals.

areas compared with other occupational areas. Age was positively associated with skin cancer incidence. The cohort effect demonstrated a decreasing rate ratio (RR) among men born after 1945, while no significant change in RR was observed among women. The period effect showed no significant influence on RR in either sex.

## Conclusions

Although the incidence of skin cancer in Songkhla, Thailand, has shown a decreasing trend in men and remained stable in women, awareness and prevention should continue to be emphasized, particularly among older individuals who are more prone to UV radiation exposure.

## 1. Introduction

Skin cancers can be divided into melanoma and nonmelanoma skin cancer (NMSC). The two primary subtypes of NMSC are basal cell carcinoma (BCC) and squamous cell carcinoma (SCC). According to GLOBOCAN 2022, non-melanoma skin cancer (excluding BCC) is the third most common cancer in men and the seventh in women, with age-adjusted incidence rates (ASRs) of 14.0 and 7.5 per 100,000, respectively. By contrast, melanoma is less prevalent, with ASRs of 3.7 and 2.9 per 100,000 in men and women, respectively [1]. The incidence of NMSC varies significantly across countries, with the highest reported in Australia [2]. In Thailand, the ASRs for melanoma and NMSC in 2016–2018 were 4.3 in women and 4.0 in men, whereas higher rates were observed in Songkhla province (4.9 in women and 4.8 in men) [3]. A rising trend in the incidence of skin cancer has been reported in various regions worldwide, including Australia, European countries, China and Hong Kong [4–7]. However, data on incidence trends in Southeast Asian countries, including Thailand, remain limited.

Ultraviolet (UV) exposure is a well-known risk factor for skin cancer [8–10]. An increasing UV index over time, due to stratospheric ozone depletion, has been correlated with the rising incidence of skin cancer in Australia [11]. Individuals engaged in outdoor occupations, such as farming and fishing, are more likely to experience higher levels of UV exposure [12]. Additionally, clothing style may influence the degree of UV exposure [13,14]. However, no data have demonstrated whether these factors directly affect the incidence of skin cancer.

Songkhla Province, located in southern Thailand, comprises 16 districts with predominant occupations varying by geographical area, such as fishing, farming, rubber plantation, and salaried employment. The religious composition also differs across regions, with both Buddhist and Muslim populations represented. These demographic and occupational variations may result in differing levels of UV exposure across districts, potentially influencing the incidence of skin cancer. Aging is another recognized factor associated with the development of skin cancer [8–10]. In Hong Kong, population aging has been identified as a contributor to the increasing trend in skin cancer incidence [7]. Thailand is similarly undergoing a demographic shift toward an aging society [15]. Therefore, population aging may also play a role in the observed trends in skin cancer incidence.

Describing the trends in skin cancer incidence based on the geographic areas and potential risk factors may provide valuable insights for healthcare planning aimed at reducing and preventing skin cancer. Thus, the present study aimed to assess trends in skin cancer incidence in Songkhla, Thailand, by age, calendar period, birth cohort, and geographic area.

## 2. Materials and methods

This study was approved by the Human Research Ethics Committee, Faculty of Medicine, Prince of Songkla University. Informed consent was not required because this was a data analysis–based study that did not involve direct contact with the participants.

### 2.1. Data source

Patient data from the Songkhla population-based cancer registry for the period 1989–2020 were obtained from the Division of Digital Innovation and Data Analytics, Faculty of Medicine, Prince of Songkla University. The data were accessed on 17 October 2025. The data that could identify the patients were not assessed. We included the records of patients diagnosed with skin cancer using the International Classification of Diseases, 10th revision, codes: C43 for malignant melanoma of the skin and C44 for other malignant neoplasms of the skin. For cases coded as C44, only SCC and BCC were included, identified using the International Classification of Diseases for Oncology, 3rd edition, codes 805–808 for squamous cell neoplasms and 809 for basal cell carcinoma. Data including age, sex, date of birth, date of diagnosis, religion (Buddhist, Muslim, and others), residential district, histologic type, tumor site (face, body, and acral area), and disease stage (local/regional/distance) were collected, all of which were recorded in cancer registry forms. Patients with no data on age and sex were excluded, as the incidence rates could not be calculated.

The Songkhla Population-Based Cancer Registry, located within the Faculty of Medicine, Prince of Songkla University, conducts active case finding through multiple data sources, including three tertiary hospitals, community hospitals, private hospitals, and the Bureau of Registration Administration, Department of Provincial Administration, Ministry of Interior [3]. For overall skin cancer, cases identified from death certificates only (DCO) accounted for 0.5% (below the 5% threshold), and microscopically verified (MV) cases represented 97.8% (above the 85% benchmark), meeting the data quality standards recommended by the International Agency for Research on Cancer (IARC) [16].

### 2.2. Statistical analysis

#### 2.2.1. ASR.
The age-specific incidence rate for each year (1989–2020) was calculated based on 18 distinct age groups (categorized in 5-year intervals, ranging from 0–4 to ≥85 years old) as the number of cases divided by the population size of that age group. These rates were subsequently adjusted to the world standard population [17] to calculate the ASRs. Each age-specific rate was multiplied by the corresponding proportion of the standard population. The sum of these values across all age groups was divided by the total standard population to obtain the ASRs, expressed per 100,000 person-years. The calculation can be expressed as:

$$\text{ASR} = \sum i\,(r_i \times w_i) / \sum i\,w_i \times 100{,}000$$

where:

$r_i$ = age-specific incidence rate in age group i

$w_i$ = number of the standard population in age group i

The population denominators for Songkhla were obtained from the Department of Public Administration, the Report of the Population Projections for Thailand 2010–2040 by the Office of the National Economic and Social Development Council [18], and the database of the Department of Provincial Administration [19].

The ASRs for all patients with skin cancer were calculated for each subgroup by sex and geographic area. The 16 districts of Songkhla Province were classified into four categories: fishing and farming areas (Ranot, Krasae Sin, Sathing Phra, and Singhanakhon), urban areas (Mueang Songkhla and Hat Yai), Muslim-predominated areas (Chana, Thepha, and Sabayoi), and rubber plantation areas (Khuan Niang, Rattaphum, Bangklam, Na Mom, Khlong Hoi Khong, Sadao, and Na Thawi), as illustrated in Fig 1. We further compared the ASRs between each category and the remaining groups, including urban versus rural, Muslim-predominated versus Buddhist-predominated, fishing and farming versus other occupations, and rubber plantation versus other occupations.

**2.2.2. Trend analysis.** *Joinpoint regression analysis*: Trends in skin cancer incidence were assessed using Joinpoint regression analysis (Joinpoint Regression Program, version 5.1.0; National Cancer Institute, Bethesda, MD, USA) to identify significant changes in trend, referred to as joinpoints. The analysis begins with the simplest model (a straight line with no joinpoints) and then tests whether adding one or more joinpoints significantly improves the fit. Each segment between joinpoints represents a time interval with its own trend, expressed as the Annual Percent Change (APCC) with 95% confidence intervals (95%CI).

*Age–period–cohort (APC) analysis*: We used APC analysis to explore how age, calendar period, and birth cohort influenced the incidence of skin cancer. In this method, incidence data are organized into 5-year age groups and 5-year calendar periods. Each case is then assigned to a corresponding birth cohort, calculated as: Birth cohort = Calendar period – Age.

We used the function *apc.fit* from the *Epi* package in R to apply a log-linear Poisson regression model and estimate incidence rates with age, period, and cohort. Because age, period, and cohort are linearly dependent, the model applies

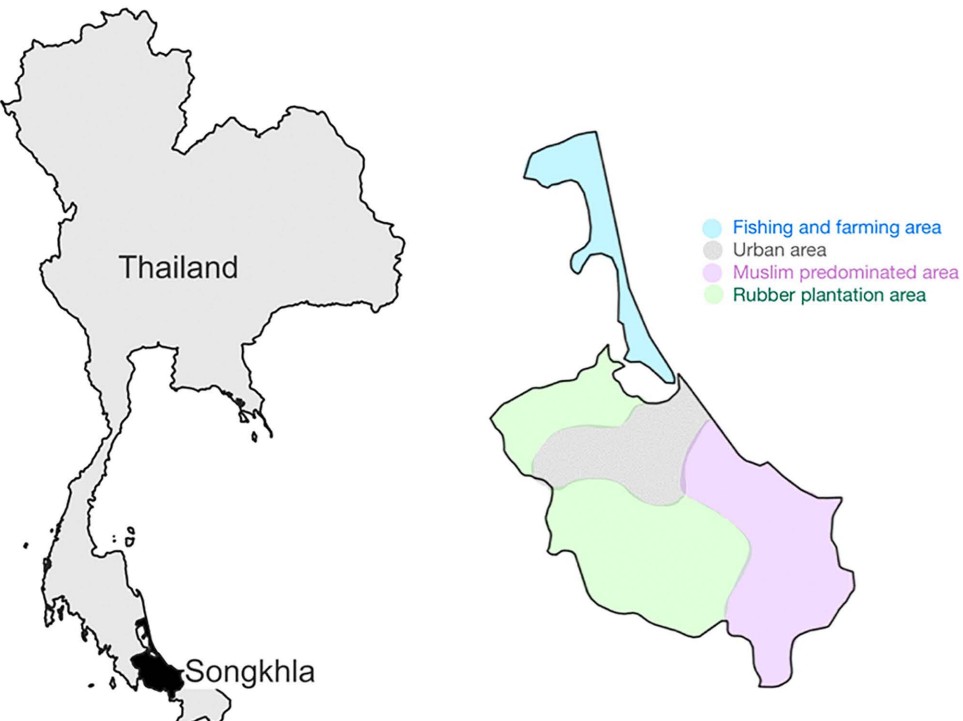

**Fig 1. Map of Songkhla province showing 4 grouped district areas.** The base map was created using public-domain administrative boundary data from Natural Earth in QGIS. Grouped district areas are shown schematically for illustrative purposes only and do not represent official administrative boundaries.

statistical constraints to separate their effects, resulting in the AP–C model and the AC–P model. In the AP–C model, the cohort effect is constrained to estimate age and period effects, while in the AC–P model, the period effect is constrained to estimate age and cohort effects. Results from both models are presented as age effects (incidence rates per 100,000), period effects and cohort effects (rate ratios (RR) with 95%CI), each compared with a reference group.

## 3. Results

### 3.1. Demographic and clinical characteristics and overall annual incidence of skin cancer

A total of 2,216 patients diagnosed with any type of skin cancer between 1989 and 2020 were retrieved. We excluded 261 cases with skin cancer types other than melanoma, SCC, or BCC, and 43 cases with missing age or sex information. No cases had missing date of diagnosis or address data. Finally, 1,912 patients were included for further analysis. The characteristics of patients with incident cases are presented in Table 1. The number of women was slightly higher than that of men. The median age at diagnosis for both sexes was 69.5 years. The most common cancer types were squamous cell carcinoma in men and basal cell carcinoma in women. The head and neck were the most common cancer sites.

The incidence of skin cancer between 1989 and 2020 is shown in Fig 2 and S1 Table, with ASRs ranging from 2.02 to 6.42 in men and from 2.01 to 5.94 in women. The ASRs was higher in men than in women in most years, except for 1989, 1991, 1992, 1996, 2014–2016, and 2018. The highest ASR was recorded in 1996 for women and in 2001 for men.

### 3.2. Incidence trend by sex and geographic area

From 1989 to 2001, the ASRs of skin cancer among men showed an increasing trend before declining (Fig 3 and S1 Table). In 1989, the ASR in men was 2.90 per 100,000 population, rising to 5.55 in 2001, with an annual percentage change (APCC) of 5.55% (95%CI 0.67, 90.50). After 2001, the ASR decreased to 3.61, with an APCC of −2.24% (95%CI −13.65, −0.42). Among women, the ASR remained relatively stable from 1989 to 2020 (3.53 − 3.81), with an APCC of 0.24% (95%CI −0.68, 1.17).

The incidence trends by sex and geographic area are shown in Fig 4 and S2 Table. Among men, the highest ASR was observed in the urban area, followed by the fishing and farming area, the rubber plantation area, and the Muslim-predominant area. Overall, the trends in all four areas showed a slight but non-significant decrease. Among women, a significant change in trend was observed in the fishing and farming area, where the incidence declined after 2016, with an APCC of −23.95% (95% CI −69.52, −0.90). No significant changes were observed in the remaining areas.

When comparing each sex and geographic area with the remaining groups (Fig 5 and S2 Table), the following patterns were observed. The urban area had a higher incidence than the rural area in both men and women. In men, the rural area showed a decreasing incidence trend after 2010 (APCC –7.40; 95%CI –40.91, –2.45), and in women, a similar decline was observed after 2016 (APCC –12.72; 95%CI –37.85, 0.02). The rubber plantation area and the Muslim-predominant area had lower incidences than other occupational areas and the Buddhist-predominant area, respectively. The Buddhist-predominant area showed a declining incidence trend after 2017 in both men and women, with APCCs of –18.46 (95%CI –42.11, –2.05) and –17.06 (95%CI –39.04, –2.18), respectively. The incidence in the fishing and farming area did not differ substantially from that in other occupational areas for either men or women.

### 3.3. Age-period-cohort analysis

Among men, both the AP–C and AC–P models demonstrated that skin cancer incidence increased with age (Fig 6A, left; S3 Table). In the AP–C model, where the cohort effect was constrained, the incidence rate ratios (RRs) did not change significantly across periods (Fig 6A, right; S4 Table). In the AC–P model, where the period effect was constrained, the incidence began to decline in the 1945–1949 birth cohort (RR 0.81; 95%CI 0.68, 0.96) and continued to decrease in subsequent cohorts, reaching 0.55 (95%CI 0.30, 0.99) in the 1975–1979 cohort (Fig 6A, middle; S5 Table).

**Table 1. Demographic and clinical characteristics of men and women patients with skin cancer in Songkhla, Thailand, 1989–2020.**

| Characteristics | Number (%) | |
|---|---|---|
| | **Men (n = 926)** | **Women (n = 986)** |
| Age (years) (median ± IQR) | 69.5 ± 19 | 69.5 ± 20 |
| **Religion** | | |
| Buddhist | 785 (84.8) | 867 (87.9) |
| Muslim | 123 (13.3) | 108 (11.0) |
| Christian | 15 (1.6) | 8 (0.8) |
| Other | 2 (0.2) | 2 (0.2) |
| Unknown | 1 (0.1) | 1 (0.1) |
| **Address** | | |
| Hatyai | 295 (31.9) | 300 (30.4) |
| Mueangsongkhla | 154 (16.6) | 169 (17.1) |
| Ranot | 69 (7.5) | 57 (5.8) |
| Sadao | 61 (6.6) | 59 (6.0) |
| Singhanakhon | 62 (6.7) | 72 (7.3) |
| Chana | 54 (5.8) | 65 (6.6) |
| Sathingphra | 51 (5.5) | 49 (5.0) |
| Rattaphum | 38 (4.1) | 49 (5.0) |
| Nathawi | 31 (3.3) | 22 (2.2) |
| Bangklam | 23 (2.5) | 27 (2.7) |
| Thepha | 24 (2.6) | 29 (2.9) |
| Khuanniang | 18 (1.9) | 32 (3.2) |
| Sabayoi | 15 (1.6) | 17 (1.7) |
| Khlonghoikhong | 12 (1.3) | 12 (1.2) |
| Krasaesin | 8 (0.9) | 14 (1.4) |
| Namom | 11 (1.2) | 13 (1.3) |
| **Cancer location** | | |
| head and neck | 402 (43.4) | 601 (61.0) |
| Lower limb and hip | 217 (23.4) | 140 (14.2) |
| Trunk | 149 (16.1) | 103 (10.4) |
| Upper limb and shoulder | 106 (11.4) | 82 (8.3) |
| Not otherwise specified | 52 (5.6) | 60 (6.1) |
| **Skin cancer type** | | |
| Squamous cell carcinoma | 491 (53.0) | 353 (35.8) |
| Basal cell carcinoma | 333 (36.0) | 537 (54.5) |
| Melanoma | 102 (11.0) | 96 (9.7) |

Among women, both models also indicated an age-related increase in skin cancer incidence (Fig 6B, left; S3 Table). However, the RRs from both models did not show significant changes (Fig 6B, middle and right; S4 and S5 Tables).

## 4. Discussion

The incidence of skin cancer in men increased until 2001 and then declined, while it remained stable among women. However, when stratified by geographic area, the incidence among women showed a decline after 2016 in rural, fishing,

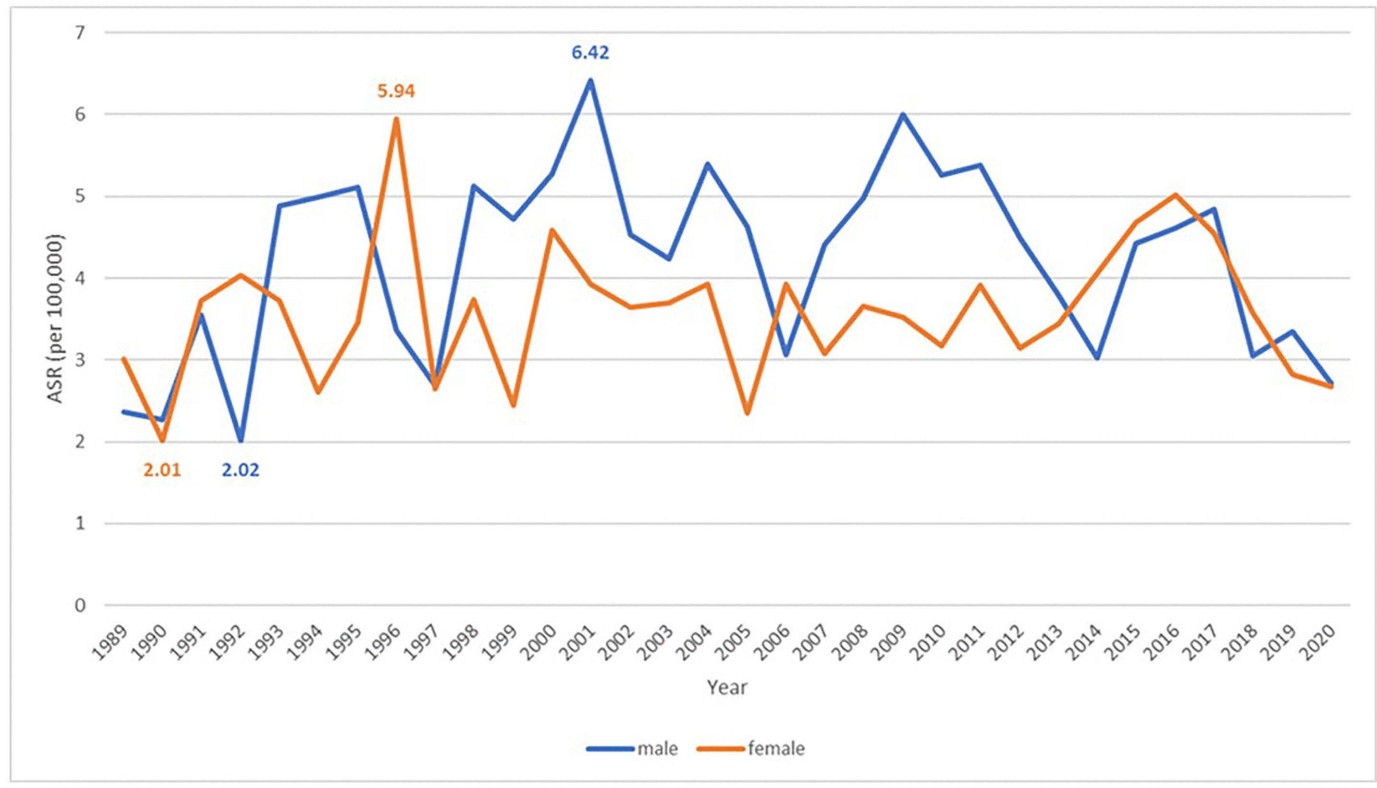

**Fig 2. Age-standardized incidence rate (ASR) of first diagnosed skin cancer per 100,000 population by sex in Songkhla, Thailand, from 1989 to 2020.**

and farming areas, as well as in predominantly Buddhist areas after 2017. Among men, the incidence also decreased in rural areas after 2010 and in Buddhist-predominated areas after 2017. Overall, incidence rates were higher in urban than in rural areas, lower in predominantly Muslim than Buddhist areas, and lower in rubber plantation areas compared with other occupational areas. Age was positively associated with skin cancer incidence. The cohort effect demonstrated a decreasing RR among men born after 1945, while no significant change in RR was observed among women. The period effect showed no significant influence on RR in either sex.

The incidence of skin cancer was slightly higher in men than in women (2.02 to 6.42 vs 2.01 to 5.94), which is consistent with the patterns reported in other countries [2]. The overall incidence trend among men increased until 2001 and subsequently declined, whereas it remained stable among women. However, a declining trend was observed in certain geographic areas among women. This pattern contrasts with trends reported in many countries, such as Western countries, Australia, China and Hong Kong, where the incidence has gradually increased [4–7]. In the United States, the incidence trends have remained relatively stable [20]. In Western countries, the increasing incidence has been attributed to the increased participation in artificial tanning and holidays in sunny climates, leading to increased exposure to UV radiation [21,22]. In some countries with a high incidence of skin cancer, such as Australia, public health campaigns have been implemented to raise awareness, promote self-examination, and encourage seeking medical attention. These efforts have contributed to increased skin cancer detection [22–24]. However, according to culture and traditions in Thailand, people prefer fair skin; therefore, they usually avoid sun exposure, and campaigns related to skin cancer are quite limited. This may be one reason why the incidence of skin cancer in Thailand has not increased.

**Multiple Joinpoint Models**

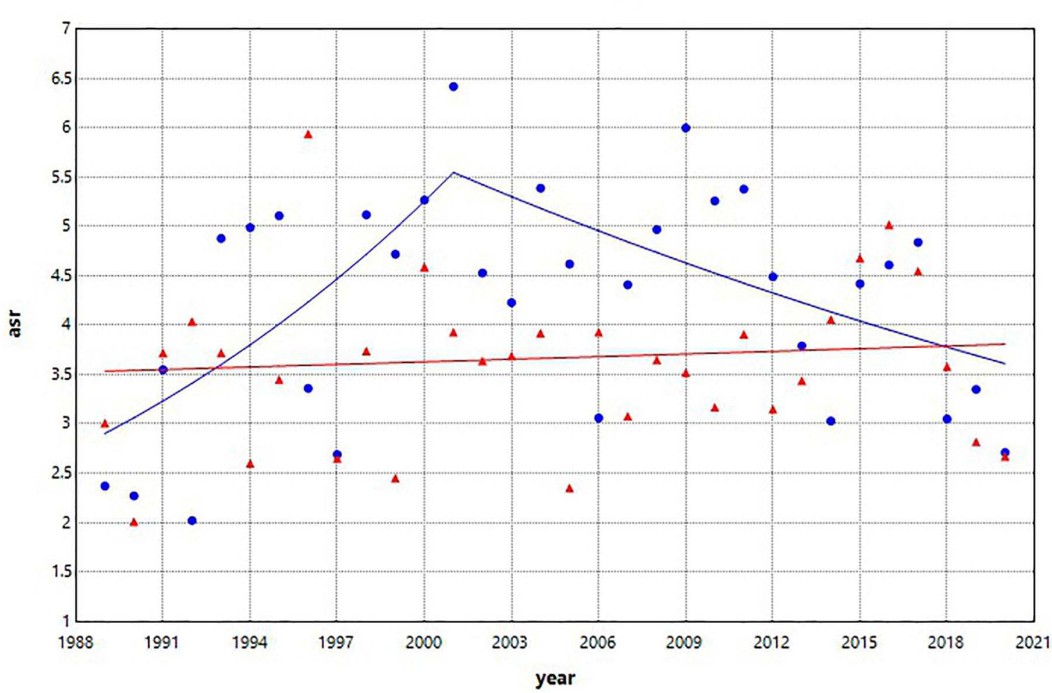

**Fig 3. Trends in skin cancer incidence by sex in Songkhla, Thailand, from 1989 to 2020, based on the Joinpoint regression analysis.**

The incidence of skin cancer varied by geographic area, with higher rates observed in urban compared to rural areas. Urban populations are generally more educated and may have greater awareness of skin cancer risk factors, more frequent use of sun protection, and a higher likelihood of noticing suspicious skin lesions as shown in the study of Nagelhout et al. [25]. In addition, the presence of three large tertiary hospitals in the urban area [3] facilitates better access to healthcare services, which likely contributes to higher detection rates. Areas with predominantly Muslim populations had lower incidence rates than those with predominantly Buddhist populations. This difference may be explained by cultural clothing practices among Muslim individuals, particularly women, who typically wear clothes that cover most of the body, thereby reducing exposure to UV radiation [13,14].

Occupational factors also influenced incidence patterns. Regions dominated by rubber plantations, where workers are often shaded during work hours, showed lower rates of skin cancer, possibly due to reduced UV exposure. In contrast, individuals residing in fishing and farming (rice field) areas typically engage in prolonged outdoor activities with greater sunlight exposure, which would be expected to result in a higher incidence of skin cancer. This expectation is supported by the study of Sripaiboonkij et al. [12], which identified occupations with high ultraviolet (UV) exposure and consequently higher skin cancer risk such as farmers, gardeners, construction workers, postal workers, park rangers, mountain guides, landscapers, and seafarers. However, our study found that the incidence of skin cancer in fishing and farming areas did not differ substantially from that in other occupational areas. This finding may be explained by protective behaviors commonly practiced in these communities, such as wearing hats and long-sleeved clothing.

According to the APC analysis, the incidence of skin cancer increased with age, which is consistent with findings from studies in China [6] and Hong Kong [7]. Aging is a well-established risk factor for skin cancer [8–10]. The cohort effect in our study demonstrated a decreasing RR among men born after 1945. This pattern is consistent with findings from China

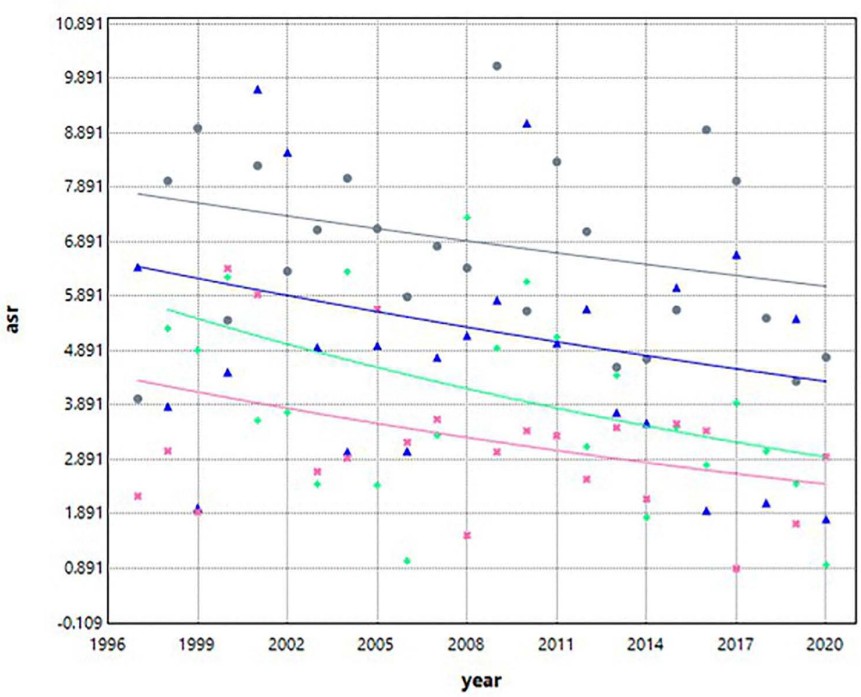

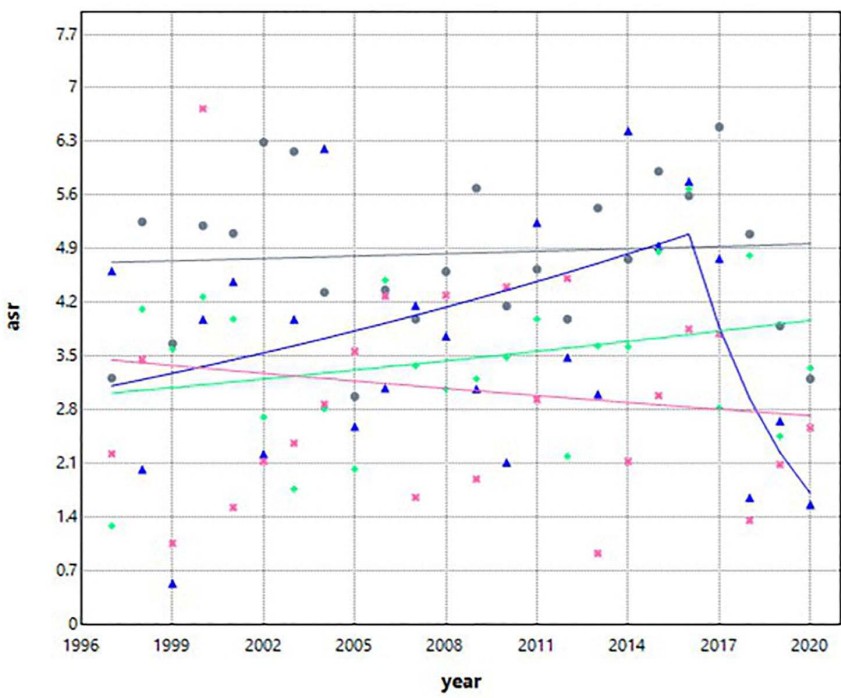

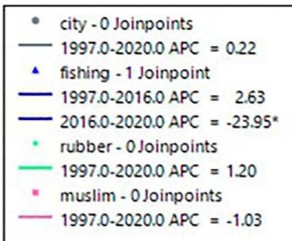

**Fig 4. Trends in skin cancer incidence by geographic area in Songkhla, Thailand, from 1989 to 2020, based on the Joinpoint regression analysis. A)** Men **B)** Women.

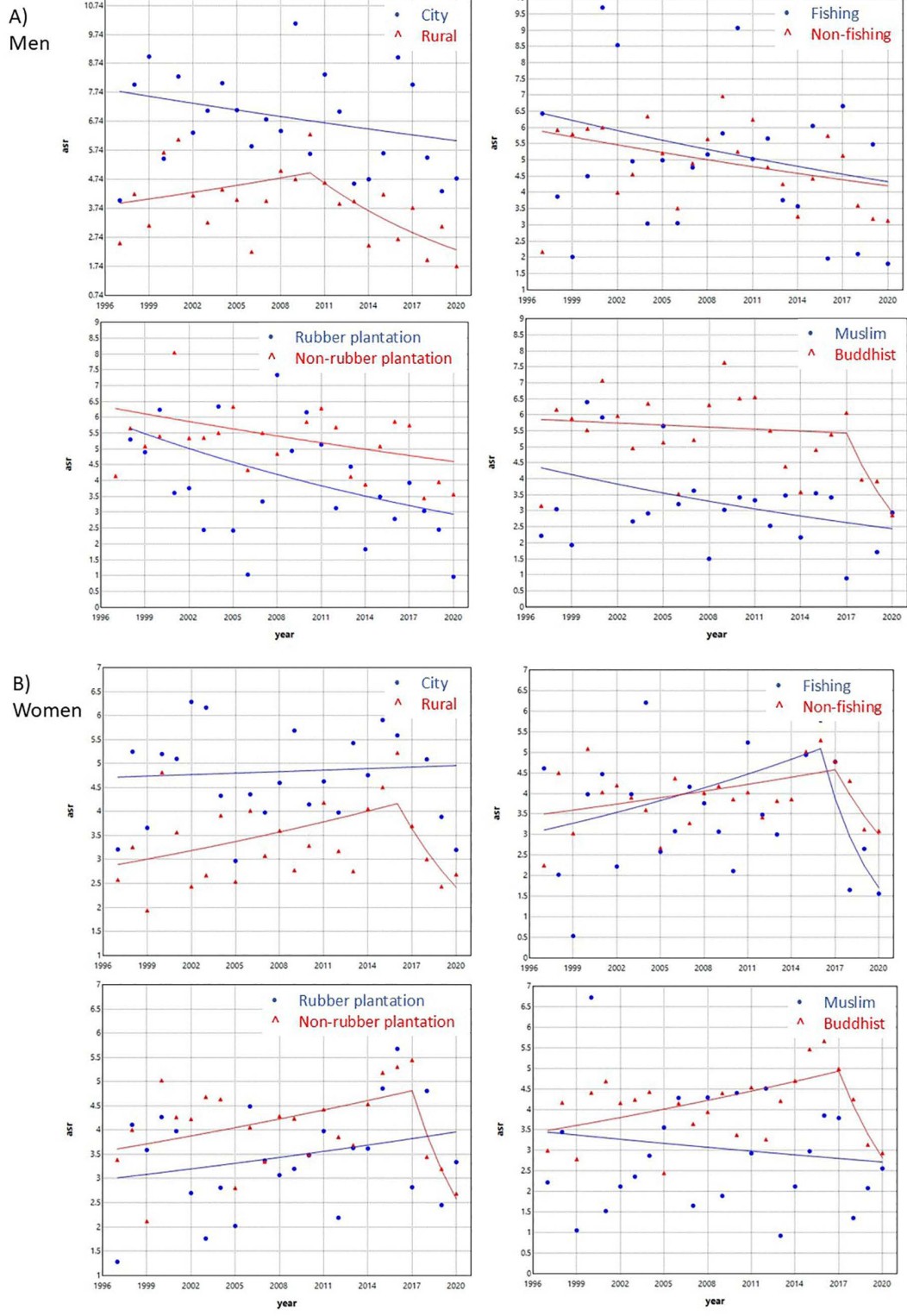

**Fig 5. Trends in skin cancer incidence by each geographic area with the remaining groups in Songkhla, Thailand, from 1989 to 2020, based on the Joinpoint regression analysis. A)** Men **B)** Women.

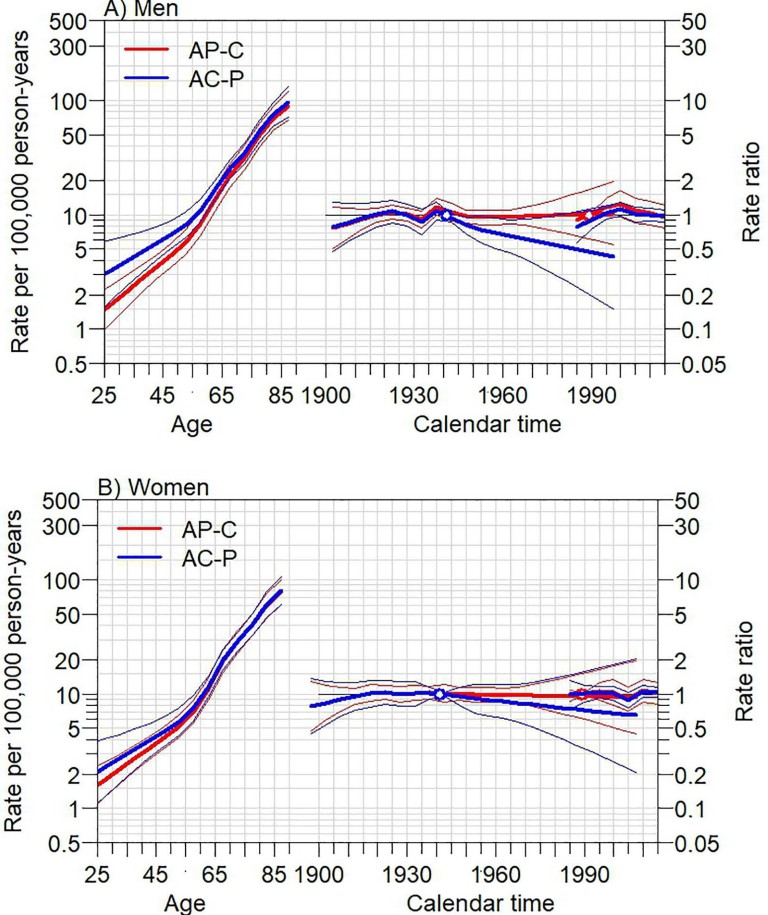

**Fig 6. Trends in skin cancer incidence in Songkhla, Thailand, from 1989 to 2020, based on Age-period-cohort analysis (AP-C and AC-P models) showing the age effect (left), cohort effect (middle), and period effect (right). A)** Men **B)** Women.

[6], where RR peaked in the 1930–1934 birth cohort, and from Hong Kong [7], where RR peaked among those born around 1975 before declining. These trends may be explained by changes in occupational patterns in Thailand. In 1960, agriculture—an occupation associated with substantial outdoor UV exposure—was the predominant form of employment, accounting for approximately 82% of the workforce. By 2008, this proportion had declined to around 35%, reflecting a shift toward industrial and service sectors [26]. In addition, sunscreen use in Thailand has increased markedly in recent decades [27,28].

The period effect in our study revealed no significant influence on RR in either sex. This contrasts with findings from China [6] and Hong Kong [7], where increasing RR over time was observed. Those studies attributed the rise to greater public awareness and improved detection of skin cancer in Hong Kong, and to increased environmental pollution in China. In Thailand, there have been no major national health policies related to skin cancer prevention, and UV radiation exposure levels have remained relatively stable. According to Bais et al. [29], UV radiation exposure in tropical regions has shown minimal variation compared to other parts of the world. Studies assessing UV radiation trends in Thailand are limited; however, one investigation conducted in Nakhon Pathom (central Thailand) [30] reported a slight increase of approximately 2% between 2001 and 2010. This change was smaller than that reported in Australia, where UV radiation increased by up to 6% per year between 1959 and 2009.

The strength of this study lies in its being among the few to examine long-term trends in skin cancer incidence using APC analysis, particularly within an Asian population. The inclusion of geographic variation further enhances the study's value by providing a context-specific understanding of disease patterns that are rarely addressed in previous research. However, some limitations should be noted. First, the dataset contained some missing information, which may have affected the results. Second, occupational and religious classifications were based on area-level rather than individual-level data, which could introduce ecological bias. Finally, as the analysis was confined to a single region in southern Thailand, it should be caution when generalizing these findings to other regions of the country or to populations with different environmental and cultural characteristics.

## 5. Conclusion

In conclusion, although the incidence of skin cancer in Songkhla, Thailand, has shown a decreasing trend in men and remained stable in women, continued emphasis on awareness and prevention is essential. Older individuals—particularly non-Muslim residents and those not engaged in rubber plantation work, who tend to experience greater outdoor UV exposure—remain at elevated risk. Strengthening public education on UV radiation, promoting sun-protective behaviors, and encouraging regular self–skin examinations are recommended strategies to further reduce the burden of skin cancer in this region.

## Supporting information

**S1 Table. Trends in skin cancer incidence by sex in Songkhla, Thailand, from 1989 to 2020, based on the Joinpoint regression analysis.**
(DOCX)

**S2 Table. Trends in skin cancer incidence by geographic areas in Songkhla, Thailand, from 1989 to 2020, based on the Joinpoint regression analysis.**
(DOCX)

**S3 Table. Age effect of incidence rates (per 100,000) in men and women in Songkhla, Thailand, from 1989 to 2020, based on the Age-period-cohort analysis (AP-C and AC-P models).**
(DOCX)

**S4 Table. Period effect of incidence rate ratios in men and women in Songkhla, Thailand, from 1985 to 2024, based on the Age-period-cohort analysis (AP-C and AC-P models).**
(DOCX)

**S5 Table. Cohort effect of incidence rate ratios in men and women in Songkhla, Thailand, from 1900 to 2009, based on the Age-period-cohort analysis (AP-C and AC-P models).**
(DOCX)

## Acknowledgments

The authors thank the staff of the Division of Digital Innovation and Data Analytics, Faculty of Medicine, Prince of Songkla University, for assistance with data collection; the Editage team for English language editing and proofreading; and colleagues from the Department of Pathology, Faculty of Medicine, Prince of Songkla University, for their valuable suggestions and encouragement.

## Author contributions

**Conceptualization:** Suchaya Pajareeyaphan, Paramee Thongsuksai.

**Data curation:** Suchaya Pajareeyaphan.

**Formal analysis:** Suchaya Pajareeyaphan.

**Investigation:** Suchaya Pajareeyaphan.

**Methodology:** Suchaya Pajareeyaphan.

**Project administration:** Suchaya Pajareeyaphan.

**Supervision:** Paramee Thongsuksai.

**Writing – original draft:** Suchaya Pajareeyaphan.

**Writing – review & editing:** Paramee Thongsuksai, Hutcha Sriplung, Wit Wichaidit.

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
