## [Decision Letter · Decision Letter 0]

15 Sep 2025

Dear Dr. Pajareeyaphan,

Thank you for submitting your manuscript to PLOS ONE. After careful consideration, we feel that it has merit but does not fully meet PLOS ONE’s publication criteria as it currently stands. Therefore, we invite you to submit a revised version of the manuscript that addresses the points raised during the review process.

We look forward to receiving your revised manuscript.

Kind regards,

Phuping Sucharitakul

Academic Editor

PLOS ONE

Journal Requirements:

5. http://creativecommons.org/licenses/by/4.0/).%20Please%20be%20aware%20that%20this%20license%20allows%20unrestricted%20use%20and%20distribution,%20even%20commercially,%20by%20third%20parties.%20Please%20reply%20and%20provide%20explicit%20written%20permission%20to%20publish%20XXX%20under%20a%20CC%20BY%20license%20and%20complete%20the%20attached%20form.”%0b%0bPlease%20upload%20the%20completed%20Content%20Permission%20Form%20or%20other%20proof%20of%20granted%20permissions%20as%20an%20%22Other%22%20file%20with%20your%20submission.%0b%0bIn%20the%20figure%20caption%20of%20the%20copyrighted%20figure,%20please%20include%20the%20following%20text:%20“Reprinted%20from%20%5bref%5d%20under%20a%20CC%20BY%20license,%20with%20permission%20from%20%5bname%20of%20publisher%5d,%20original%20copyright%20%5boriginal%20copyright%20year%5d.”%0b%0bb.%20If%20you%20are%20unable%20to%20obtain%20permission%20from%20the%20original%20copyright%20holder%20to%20publish%20these%20figures%20under%20the%20CC%20BY%204.0%20license%20or%20if%20the%20copyright%20holder’s%20requirements%20are%20incompatible%20with%20the%20CC%20BY%204.0%20license,%20please%20either%20i)%20remove%20the%20figure%20or%20ii)%20supply%20a%20replacement%20figure%20that%20complies%20with%20the%20CC%20BY%204.0%20license.%20Please%20check%20copyright%20information%20on%20all%20replacement%20figures%20and%20update%20the%20figure%20caption%20with%20source%20information.%20If%20applicable,%20please%20specify%20in%20the%20figure%20caption%20text%20when%20a%20figure%20is%20similar%20but%20not%20identical%20to%20the%20original%20image%20and%20is%20therefore%20for%20illustrative%20purposes%20only.%0bThe%20following%20resources%20for%20replacing%20copyrighted%20map%20figures%20may%20be%20helpful:%0b%0bUSGS%20National%20Map%20Viewer%20(public%20domain):%20http://viewer.nationalmap.gov/viewer/%0bThe%20Gateway%20to%20Astronaut%20Photography%20of%20Earth%20(public%20domain):%20http://eol.jsc.nasa.gov/sseop/clickmap/%0bMaps%20at%20the%20CIA%20(public%20domain):%20https://www.cia.gov/library/publications/the-world-factbook/index.html%20and%20https://www.cia.gov/library/publications/cia-maps-publications/index.html%0bNASA%20Earth%20Observatory%20(public%20domain):%20http://earthobservatory.nasa.gov/%0bLandsat:%20http://landsat.visibleearth.nasa.gov/%0bUSGS%20EROS%20(Earth%20Resources%20Observatory%20and%20Science%20(EROS)%20Center)%20(public%20domain):%20http://eros.usgs.gov/# Natural Earth (public domain): http://www.naturalearthdata.com/ " xlink:type="simple">We note that Figure 1 in your submission contain map/satellite images which may be copyrighted. All PLOS content is published under the Creative Commons Attribution License (CC BY 4.0), which means that the manuscript, images, and Supporting Information files will be freely available online, and any third party is permitted to access, download, copy, distribute, and use these materials in any way, even commercially, with proper attribution. For these reasons, we cannot publish previously copyrighted maps or satellite images created using proprietary data, such as Google software (Google Maps, Street View, and Earth). For more information, see our copyright guidelines: http://journals.plos.org/plosone/s/licenses-and-copyright.

Reviewers' comments:

Reviewer's Responses to Questions

**Comments to the Author**

1. Is the manuscript technically sound, and do the data support the conclusions?

Reviewer #1: Partly

Reviewer #2: Partly

2. Has the statistical analysis been performed appropriately and rigorously?

Reviewer #1: No

Reviewer #2: Yes

3. Have the authors made all data underlying the findings in their manuscript fully available?

Reviewer #1: No

Reviewer #2: Yes

4. Is the manuscript presented in an intelligible fashion and written in standard English?

Reviewer #1: Yes

Reviewer #2: Yes

Reviewer #1: This article reported escribing the trends in skin cancer incidence based on demographic and occupational variations may result in differing levels of UV exposure across districts, potentially influencing the incidence of skin cancer.

There are some points that need to be modified. Please find some comments below:

Title

The authors should remove the word “geographic variation” from the title because there is no spatial analysis in this study.

I would suggest the title as follow: “Trends in skin cancer incidence in Songkhla, Southern Thailand, 1989–2020: A population-base study on impact of geographic variation”

Abstract

1) Is the incident presented per 100000? Please clarify.

Introduction

1) Please add the references showing that “Individuals engaged in outdoor occupations, such as farming and fishing, are more likely to experience higher levels of UV exposure. Additionally, clothing style may influence the degree of UV exposure.”

Method

1) How many case were excluded because data on date of diagnosis, age, and sex are missing? How the authors manage when the address of the patient is missing at district level?

2) It would be interesting to see the comparison of the trend incidence in city vs rural district, Muslim vs.non Muslim predominated area, and farming, fishing, gardening and other occupations area. The author cannot conclude the impact of these factors without the control group even though the significant change of incidences are shown because it might also change in the control groups.

3) The change of levels of UV exposure across districts might potentially influence the incidence of skin cancer. It would be interesting if the author could add the information of UV exposure over time in the manuscript.

4) As shown in Figure 1, the author simply classified the 16 districts of Songkhla were grouped into 4 category 1) farming and fishing areas (Ranot, Krasae Sin, Sathing Phra, and Singhanakhon), 2) city areas (Mueang Songkhla and Hat Yai), 3) Muslim predominated areas (Chana, Thepha, and Sabayoi), and 4) gardening areas (Khuan Niang, Rattaphum, Bangklam, Na Mom, Khlong Hoi Khong, Sadao, and Na Thawi) but this classification pool the impact of urbanization, occupation and region together. I would suggest the authors re-classify the group as city (urban) vs rural district, Muslim vs. non-Muslim predominated area, and farming, fishing, gardening and other occupations area.

Results

1) Please clarify this sentence. What are these figures?

“Among the male population, the highest ASR was observed in the city area (3.10–7.14), followed by the fishing and farming area (3.67–6.42), the gardening area (1.18–4.90), and the Muslim predominated area (2.24– 3.93).”

2) Data quality over time of cancer registry is important in particular in the trend analysis. I would suggest authors to add the table of data quality in the results part or as the supplementary file.

Discussion

1) The authors should add the references to support the following statement.

“These variations suggest that the occupational environment and clothing practices may have affected the incidence. Muslim individuals, particularly women, typically wear clothing that covers most of the body, resulting in reduced exposure to UV radiation. The gardening area in Songkhla primarily consists of rubber plantations, which provide shade during working and may reduce exposure to UV radiation. By contrast, individuals residing in fishing and farming areas—particularly those working in rice fields—are generally exposed to greater amounts of sunlight, resulting in a higher incidence of skin cancer. The highest incidence was observed in the city area. This may be attributed to multiple factors. Urban populations are generally more highly educated. Moreover, the presence of three large tertiary hospitals in the city area facilitates better access to healthcare services.”

2) Please add strength and limitation of the study in the discussion part.

Conclusion

The conclusion needs to be modified according to the new analysis I proposed.

Minor:

1) Please make sure that all the abbreviations are described.

2) Has this article been written using AI? If so, please provide the reference appropriately.

3) Please complete the declaration part including Acknowledgement, Funding, Declarations of interest, Author contributions, and Data availability statement.

Reviewer #2: Trends and geographic variation in skin cancer incidence in Songkhla, Southern Thailand, 1989–2020: A Joinpoint regression and age-period-cohort analysis

The authors presented an interesting approach to skin cancer data from Songkhla (Thailand) population-based registry. They aimed to obtain insights on skin cancer incidence based on an ecological approach. The statistical approach is appropriated; they conducted joinpoint regression analyses, including age-period-cohort analysis. Although cancer time trend analysis has been extensively explored in a large number of papers, research from Thailand is scarce.

Authors sought Ethics approval from an Institutional Review Board

Abstract conveys the main results of this research

However, some points deserve particular attention and should be explained in more detail. The following are the comments/suggestions:

Abstract

The conclusion should be focused on practical implications of these results. The authors should avoid to repeat results in the conclusion subheading

Methods

1– The world standardized population used by the authors demands a reference.

2 - The authors used the same acronym – APC – for “annual percent change” and “age period cohort”. They should modify one of these acronyms

3 – Statistical approach should be described in deep. For readers not familiarized with joinpoint regression model, an explanation in brief of such approach would be welcome. Also, it tis not clear for me how did authors performed age-period-cohort analysis. I suppose the authors assumed a multiple regression model with a Poisson distribution. However, there is no explanation about this regression model. Indeed the statistical approach subheading is poor and should be rewritten. Did authors use de joinpoint regression software for age-period-cohort analysis? Based on figures, I supose that yes, but there is no information about this. Also, authors should explain what measures they obtained from these regression models. For example, anual percent change, relative risk for age, period, and cohort. Confidence Intervals for such measures are useful and should be provided.

Results

4 – Figures with results should be improved, it is very difficult to read results from these figures, particularly age-period-cohort results.

5 - Although the authors described in detail time trends results, there is no description about age-period-cohort results. They did not present results on relative risk for age, period, and cohort. A table with these results would make easier to read them.

Discussion

6 - The authors should avoid to repeat results in this section.

7 – In the second paragraph, the statements such as: “In Western countries the increasing incidence has been…” or “In some countries with a high incidence…” demand bibliographic references.

8 – In the fourth paragraph, the authors wrote “Urban populations are generally more highly educated.” This statement demands further explanation on how this characteristic play a role in the occurrence of higher incidence rate. Do more educated people use more frequenly health services or cancer screening? However, more educated people also tend to be aware about risk factors or cancer prevention. The explanation that authors presented is not enough, they should discuss these results in more detail.

9 – The authors should present and discuss the limitations of this study. Please remember, this study was based on an ecological approach.

Conclusion

10 – The conclusion is not supported by the results. The authors stated that the decline in incidence rates was explained by occupational patterns and cultural and tradicional practices. However, the decrease in incidence rates was observed in all groups, this means that occupation, culture or traditional practices could explain variation between areas, but they can’t explain the decrease in incidence rates observed. The explanation of this decrease should be based on factors shared by all groups.

11 - The conclusion should be focused on practical implications of these results.

**Do you want your identity to be public for this peer review?** For information about this choice, including consent withdrawal, please see our Privacy Policy

Reviewer #1: No

Reviewer #2: **Yes:**  Cristina Teixeira

---

## [Author Response · Author response to Decision Letter 1]

29 Oct 2025

29/10/25

Emily Chenette

Editor-in-Chief

PLOS ONE

Dear Editor:

This letter accompanies the revised version of our manuscript entitled “Trends in Skin Cancer Incidence in Songkhla, Southern Thailand, 1989–2020: A Population-Based Study on the Impact of Geographic Variation” (PONE-D-25-44756), submitted for publication in PLOS ONE.

We appreciate the valuable comments and suggestions provided by the Associate Editor and the reviewers, which have greatly helped us improve the quality of our paper. All revisions have been marked using track changes.

Below, we provide a summary of the modifications made in the manuscript to address the reviewers’ comments.

We hope that our revisions and responses satisfactorily address all concerns and that the revised manuscript will be suitable for publication in PLOS ONE.

Sincerely,

Suchaya Pajareeyaphan, M.D.

Point-by-point responses to reviewers

Journal Requirements:

Response: We have carefully reviewed and revised our manuscript to ensure full compliance with PLOS ONE’s style and formatting requirements, including file naming conventions.

Response: We have now included the numerical values used to generate the graphs in the Supporting Information files. However, we are unable to provide more detailed data because the dataset was obtained from the Division of Digital Innovation and Data Analytics, Faculty of Medicine, Prince of Songkla University. Due to institutional and patient privacy restrictions, these data are not publicly available but may be made available from the corresponding author upon reasonable request and with permission from the Human Research Ethics Committee, Faculty of Medicine, Prince of Songkla University.

3. When completing the data availability statement of the submission form, you indicated that you will make your data available on acceptance.

Response: We have added the Data Availability Statement to the manuscript (page 12, lines 9–12).

4. Please include your full ethics statement in the ‘Methods’ section of your manuscript file.

Response: We have included the complete ethics statement in the ‘Methods’ section (page 3, lines 33–36).

5. We note that Figure 1 in your submission contain map/satellite images which may be copyrighted.

Response: Figure 1 was created entirely by the authors and does not contain any copyrighted material. The map was self-generated for illustrative purposes using original data and design.

Reviewer #1: This article reported describing the trends in skin cancer incidence based on demographic and occupational variations may result in differing levels of UV exposure across districts, potentially influencing the incidence of skin cancer.

There are some points that need to be modified. Please find some comments below:

Title

The authors should remove the word “geographic variation” from the title because there is no spatial analysis in this study.

I would suggest the title as follow: “Trends in skin cancer incidence in Songkhla, Southern Thailand, 1989–2020: A population-base study on impact of geographic variation”

Response: We have revised the title according to your suggestion.

Abstract

1) Is the incident presented per 100000? Please clarify.

Response: Yes, the incidence is presented per 100,000 population. We have clarified this in the Abstract (page 2, line 13).

Introduction

1) Please add the references showing that “Individuals engaged in outdoor occupations, such as farming and fishing, are more likely to experience higher levels of UV exposure. Additionally, clothing style may influence the degree of UV exposure.”

Response: We have added the appropriate references to support this statement (page 3, lines 15–17).

Method

1) How many case were excluded because data on date of diagnosis, age, and sex are missing? How the authors manage when the address of the patient is missing at district level?

Response: Initially, 333 cases had missing diagnosis dates. After consulting with the data provider, we discovered that these missing values were due to technical issues during the initial data retrieval process. The data provider subsequently re-extracted the dataset to correct this problem, resulting in an updated dataset. The revised dataset included 2,216 patients diagnosed with any type of skin cancer between 1989 and 2020. We excluded 261 cases with skin cancer types other than melanoma, SCC, or BCC, and 43 cases with missing age or sex information. No cases had missing data for diagnosis date or address. Consequently, 1,912 patients were included in the final analysis. These details have been added to the Results section (page 6, lines 5–8). The results from the new dataset showed some changes, with a decreasing trend in men and a stable trend in women.

2) It would be interesting to see the comparison of the trend incidence in city vs rural district, Muslim vs.non Muslim predominated area, and farming, fishing, gardening and other occupations area. The author cannot conclude the impact of these factors without the control group even though the significant change of incidences are shown because it might also change in the control groups.

Response: Thank you for this valuable suggestion. We have added comparative analyses for urban vs. rural areas, Muslim-predominated vs. Buddhist-predominated areas, fishing and farming areas vs. other occupational areas, and rubber plantation areas (previously referred to as gardening areas) vs. other occupational areas. We did not combine fishing and farming with gardening because, in Songkhla, “gardening” primarily refers to rubber plantations, where workers experience different levels of UV exposure. The results are presented on page 8, lines 22–30, Figure 5, and Table S2. We have also replaced “gardening areas” with “rubber plantation areas” to more accurately reflect the occupational environment and its UV exposure characteristics.

3) The change of levels of UV exposure across districts might potentially influence the incidence of skin cancer. It would be interesting if the author could add the information of UV exposure over time in the manuscript.

Response: Studies on long-term UV radiation trends in Thailand are limited. However, we have added a reference to one study conducted in Nakhon Pathom (Central Thailand) in the Discussion section (page 10, lines 37–38).

4) As shown in Figure 1, the author simply classified the 16 districts of Songkhla were grouped into 4 category 1) farming and fishing areas (Ranot, Krasae Sin, Sathing Phra, and Singhanakhon), 2) city areas (Mueang Songkhla and Hat Yai), 3) Muslim predominated areas (Chana, Thepha, and Sabayoi), and 4) gardening areas (Khuan Niang, Rattaphum, Bangklam, Na Mom, Khlong Hoi Khong, Sadao, and Na Thawi) but this classification pool the impact of urbanization, occupation and region together. I would suggest the authors re-classify the group as city (urban) vs rural district, Muslim vs. non-Muslim predominated area, and farming, fishing, gardening and other occupations area.

Response: We have reclassified the areas according to your suggestion and incorporated the revised analysis as described in 2).

Results

1) Please clarify this sentence. What are these figures?

“Among the male population, the highest ASR was observed in the city area (3.10–7.14), followed by the fishing and farming area (3.67–6.42), the gardening area (1.18–4.90), and the Muslim predominated area (2.24– 3.93).”

Response: We have deleted this sentence due to data reanalysis and replaced it with a clearer description based on the updated results (page 8, lines 12–17; Figure 4; S2 Table).

2) Data quality over time of cancer registry is important in particular in the trend analysis. I would suggest authors to add the table of data quality in the results part or as the supplementary file.

Response: We have added details regarding data quality assessment in the Methods section (page 4, lines 14–20).

Discussion

1) The authors should add the references to support the following statement.

“These variations suggest that the occupational environment and clothing practices may have affected the incidence. Muslim individuals, particularly women, typically wear clothing that covers most of the body, resulting in reduced exposure to UV radiation. The gardening area in Songkhla primarily consists of rubber plantations, which provide shade during working and may reduce exposure to UV radiation. By contrast, individuals residing in fishing and farming areas—particularly those working in rice fields—are generally exposed to greater amounts of sunlight, resulting in a higher incidence of skin cancer. The highest incidence was observed in the city area. This may be attributed to multiple factors. Urban populations are generally more highly educated. Moreover, the presence of three large tertiary hospitals in the city area facilitates better access to healthcare services.”

Response: We have extensively revised the Discussion to align with the updated results and added supporting references as suggested.

2) Please add strength and limitation of the study in the discussion part.

Response: We have added a paragraph discussing the strengths and limitations of our study at the end of the Discussion (page 10, line 41–page 11, line 7).

Conclusion

The conclusion needs to be modified according to the new analysis I proposed.

Response: We have revised the Conclusion in both the Abstract and the main text to reflect the updated analyses (page 11, lines 9–12).

Minor:

1) Please make sure that all the abbreviations are described.

Response: All abbreviations are now defined at first mention and summarized on page 1.

2) Has this article been written using AI? If so, please provide the reference appropriately.

Response: We have added an “AI Use Disclosure” section in the manuscript (page 11, lines 20–23).

3) Please complete the declaration part including Acknowledgement, Funding, Declarations of interest, Author contributions, and Data availability statement.

Response: All declaration sections have been completed (page 11, lines 4–page 12, lines 14).

Reviewer #2: Trends and geographic variation in skin cancer incidence in Songkhla, Southern Thailand, 1989–2020: A Joinpoint regression and age-period-cohort analysis

The authors presented an interesting approach to skin cancer data from Songkhla (Thailand) population-based registry. They aimed to obtain insights on skin cancer incidence based on an ecological approach. The statistical approach is appropriated; they conducted joinpoint regression analyses, including age-period-cohort analysis. Although cancer time trend analysis has been extensively explored in a large number of papers, research from Thailand is scarce.

Authors sought Ethics approval from an Institutional Review Board

Abstract conveys the main results of this research

However, some points deserve particular attention and should be explained in more detail. The following are the comments/suggestions:

Abstract

The conclusion should be focused on practical implications of these results. The authors should avoid to repeat results in the conclusion subheading

Response: We have revised the conclusion in the Abstract to highlight practical implications, as suggested.

Methods

1– The world standardized population used by the authors demands a reference.

Response: We have added the appropriate reference for the world standardized population (page 4, line 28).

2 - The authors used the same acronym – APC – for “annual percent change” and “age period cohort”. They should modify one of these acronyms

Response: We have revised the abbreviation for “annual percent change” to “APCC.”

3 – Statistical approach should be described in deep. For readers not familiarized with joinpoint regression model, an explanation in brief of such approach would be welcome. Also, it tis not clear for me how did authors performed age-period-cohort analysis. I suppose the authors assumed a multiple regression model with a Poisson distribution. However, there is no explanation about this regression model. Indeed the statistical approach subheading is poor and should be rewritten. Did authors use de joinpoint regression software for age-period-cohort analysis? Based on figures, I supose that yes, but there is no information about this. Also, authors should explain what measures they obtained from these regression models. For example, anual percent change, relative risk for age, period, and cohort. Confidence Intervals for such measures are useful and should be provided.

Response: We have substantially revised the “Statistical Analysis” section to include detailed explanations of the joinpoint regression and age–period–cohort models (page 4, lines 24–page 5, lines 33). We have also provided 95% confidence intervals for all reported values.

Results

4 – Figures with results should be improved, it is very difficult to read results from these figures, particularly age-period-cohort results.

Response: We have improved some figures and added corresponding supplementary tables for better interpretation (S1–S5 Tables).

5 - Although the authors described in detail time trends results, there is no description about age-period-cohort results. They did not present results on relative risk for age, period, and cohort. A table with these results would make easier to read them.

Response: We have added a detailed description of the age–period–cohort results (page 9, lines 1–14) and provided tables summarizing the model outputs (S3–S5 Tables).

Discussion

6 - The authors should avoid to repeat results in this section.

Response: We have revised the Discussion to reduce redundancy.

7 – In the second paragraph, the statements such as: “In Western countries the increasing incidence has been…” or “In some countries with a high incidence…” demand bibliographic references.

Response: We have added the appropriate references (page 9, lines 32–36).

8 – In the fourth paragraph, the authors wrote “Urban populations are generally more highly educated.” This statement demands further explanation on how this characteristic play a role in the occurrence of higher incidence rate. Do more educated people use more frequenly health services or cancer screening? However, more educated people also tend to be aware about risk factors or cancer prevention. The explanation that authors presented is not enough, they should discuss these results in more detail.

Response: Thank you for your suggestion. We added the explanation in more detail (page 10, line 1-6)

9 – The authors should present and discuss the limitations of this study. Please remember, this study was based on an ecological approach.

Response: We have added a section on study limitations in the Discussion (page 11, lines 2–7).

Conclusion

10 – The conclusion is not supported by the results. The authors stated that the decline in incidence rates was explained by occupational patterns and cultural and tradicional practices. However, the decrease in incidence rates was observed in all groups, this means that occupation, culture or traditional practices could explain variation between areas, but they can’t explain the decrease in incidence rates observed. The explanation of this decrease should be based on factors shared by all groups.

11 - The conclusion should be focused on practical implications of these results.

Response: We have revised the Conclusion to ensure it is fully supported by the results and emphasizes practical implications, as suggested.

---

## [Decision Letter · Decision Letter 1]

24 Nov 2025

Trends in Skin Cancer Incidence in Songkhla, Southern Thailand, 1989–2020: A Population-Based Study on the Impact of Geographic Variation

PLOS ONE

Dear Dr. Pajareeyaphan,

Thank you for submitting your manuscript to PLOS ONE. After careful consideration, we feel that it has merit but does not fully meet PLOS ONE’s publication criteria as it currently stands. Therefore, we invite you to submit a revised version of the manuscript that addresses the points raised during the review process

We look forward to receiving your revised manuscript.

Kind regards,

Phuping Sucharitakul

Academic Editor

PLOS ONE

Journal Requirements:

Reviewers' comments:

Reviewer's Responses to Questions

**Comments to the Author**

Reviewer #1: All comments have been addressed

Reviewer #2: All comments have been addressed

2. Is the manuscript technically sound, and do the data support the conclusions?

Reviewer #1: Yes

Reviewer #2: Yes

3. Has the statistical analysis been performed appropriately and rigorously?

Reviewer #1: Yes

Reviewer #2: Yes

4. Have the authors made all data underlying the findings in their manuscript fully available?

Reviewer #1: Yes

Reviewer #2: Yes

5. Is the manuscript presented in an intelligible fashion and written in standard English?

Reviewer #1: Yes

Reviewer #2: Yes

Reviewer #1: Just want to recommend the author to enrich the conclusion, pointing out the impact of geographical and occupational factors in this part and what are the suggestions to prevent or manage the skin cancer based on the results of your study.

Reviewer #2: The authors addressed all my comments and suggestions.

I thhink the manuscript was substantially improved.

I have no more comments.

**Do you want your identity to be public for this peer review?** For information about this choice, including consent withdrawal, please see our Privacy Policy

Reviewer #1: No

Reviewer #2: **Yes:**  Cristina Teixeira

---

## [Author Response · Author response to Decision Letter 2]

3 Dec 2025

Journal Requirements:

Response: The reviewer did not request citation of any specific articles, and no additional references were deemed necessary based on our evaluation. Therefore, no new citations have been added.

Response: We thoroughly re-examined our reference list to ensure accuracy and completeness. All cited works were checked against PubMed and publisher databases for retraction status. None of the references in our manuscript were found to be retracted, and thus no removals or replacements were required. Accordingly, the reference list remains unchanged. We have ensured that all entries are formatted correctly according to the journal’s style and reflect the most up-to-date publication information.

Reviewer #1: Just want to recommend the author to enrich the conclusion, pointing out the impact of geographical and occupational factors in this part and what are the suggestions to prevent or manage the skin cancer based on the results of your study.

Response: We appreciate your suggestion. We have expanded and refined the conclusion to highlight the role of geographical and occupational factors and also added specific recommendations for prevention and management of the skin cancer. These revisions improve the clarity and practical relevance of the conclusion and are now reflected in the revised manuscript (Conclusion section, page 11, lines 10-16).

---

## [Editor Report · Decision Letter 2]

10 Dec 2025

Trends in Skin Cancer Incidence in Songkhla, Southern Thailand, 1989–2020: A Population-Based Study on the Impact of Geographic Variation

PONE-D-25-44756R2

Dear Dr. Pajareeyaphan,

We’re pleased to inform you that your manuscript has been judged scientifically suitable for publication and will be formally accepted for publication once it meets all outstanding technical requirements.

Kind regards,

Dr. Phuping Sucharitakul

Academic Editor

PLOS One
---

## [Editor Report · Acceptance letter]

PONE-D-25-44756R2

PLOS One

Dear Dr. Pajareeyaphan,

I'm pleased to inform you that your manuscript has been deemed suitable for publication in PLOS One. Congratulations! Your manuscript is now being handed over to our production team.

Kind regards,

on behalf of

Dr. Phuping Sucharitakul

Academic Editor

PLOS One